# Unifying Voxel-based Representation with Transformer for 3D Object Detection

**Yanwei Li**[1]   **Yilun Chen**[1]   **Xiaojuan Qi**[2]   **Zeming Li**[3]   **Jian Sun**[3]   **Jiaya Jia**[1,4]

The Chinese University of Hong Kong[1]   The University of Hong Kong[2]
MEGVII Technology[3]   SmartMore[4]

## Abstract

In this work, we present a unified framework for multi-modality 3D object detection, named UVTR. The proposed method aims to unify multi-modality representations in the voxel space for accurate and robust single- or cross-modality 3D detection. To this end, the modality-specific space is first designed to represent different inputs in the voxel feature space. Different from previous work, our approach preserves the voxel space without height compression to alleviate semantic ambiguity and enable spatial connections. To make full use of the inputs from different sensors, the cross-modality interaction is then proposed, including knowledge transfer and modality fusion. In this way, geometry-aware expressions in point clouds and context-rich features in images are well utilized for better performance and robustness. The transformer decoder is applied to efficiently sample features from the unified space with learnable positions, which facilitates object-level interactions. In general, UVTR presents an early attempt to represent different modalities in a unified framework. It surpasses previous work in single- or multi-modality entries. The proposed method achieves leading performance in the nuScenes *test* set for both object detection and the following object tracking task. Code is made publicly available at https://github.com/dvlab-research/UVTR.

## 1   Introduction

Detecting 3D objects with multi-modality sensors (*i.e.,* LiDAR and camera) is regarded as a fundamental task in real-world scenes. For accurate object detection, data from different modalities are utilized to provide complementary knowledge, like accurate positions from point clouds and rich context from images. Toward this purpose, a unified representation is essential to facilitate knowledge transfer and feature fusion across modalities. However, due to the lack of accurate depth from cameras, images can not be naturally represented in voxel space like that of point clouds.

In the unified progress, several representations have been studied that can be roughly separated into input- and feature-level streams. For the first one, multi-modality data is aligned at the beginning of network. In particular, pseudo point clouds in Figure 1a are transformed from image aided by predicted depth [1, 2], while the range-view image in Figure 1b is projected from point clouds [3, 4]. Because of inaccurate depth in pseudo point clouds and collapsed 3D geometry in range-view images, the spatial structure of data is damaged, which brings inferior results. For feature-level method, a typical approach is to transform image features as frustum and then compress to BEV space [5, 6], like that in Figure 1c. However, due to ray-like trajectories in the frustum [7], height compression at each position aggregates features from various objects and thus introduces semantic ambiguity. Meanwhile, other implicit manners in contemporary work [8, 9, 10] can hardly support explicit feature interactions in 3D space and restrict further knowledge transfer. Therefore, a more unified representation is desired to bridge modality gap and facilitate interactions from multiple aspects.

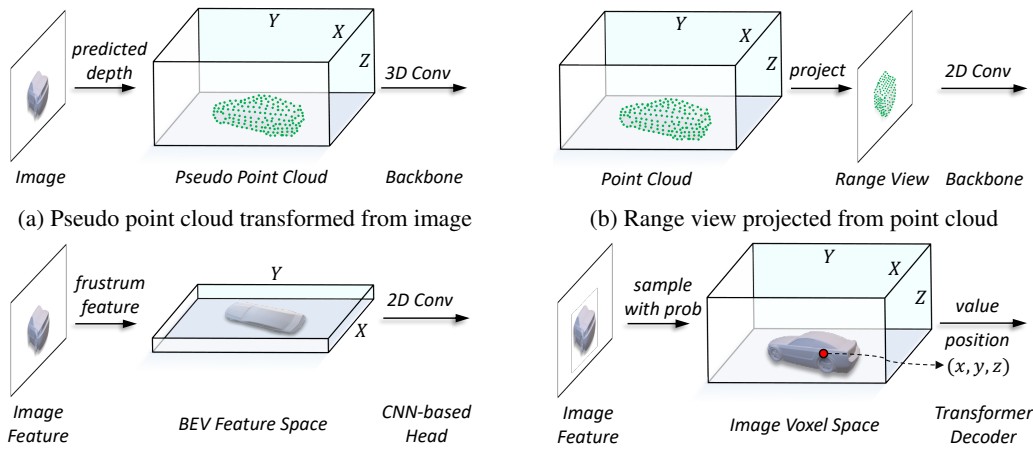

(a) Pseudo point cloud transformed from image

(b) Range view projected from point cloud

(c) BEV feature space transformed from image feature

(d) Voxel feature space sampled from image feature

Figure 1: Toy example of methods for unified representation. Compared with others, the proposed manner in 1d constructs the voxel space by sampling features from the image plane and represents multi-modalities uniformly *without* height-level compression in 1c that brings semantic ambiguity.

In this paper, we present a simple yet effective framework to unify the voxel-based representation with transformer, called UVTR. In particular, features from images and point clouds are represented and interacted in the explicit voxel-based space. For images, we construct the voxel space by sampling features from the image plane according to predicted depth scores and geometric constraints, as briefly depicted in Figure 1d. For point clouds, the accurate position naturally allows us to associate features with voxels. Then, voxel encoder is introduced for spatial interaction that establishes the relationship among adjacent features. In this way, cross-modality interaction is naturally conducted with features in each voxel space. For object-level interaction, deformable transformer [11] is adopted as the decoder that samples specific feature for each object query with position $(x, y, z)$ in the unified voxel space, as illustrated in Figure 1d. Meanwhile, the introduction of 3D query position efficiently alleviates the semantic ambiguity brought by height compression in BEV space as analysed before.

Compared with previous and even concurrent studies [8, 9], more key advances can be achieved with the proposed framework. *First*, the explicit voxel-based representation supports spatial interaction in 3D space and multi-frame scenes that bring significant improvements. *Second*, the proposed unified manner facilitates cross-modality learning and can be naturally applied for knowledge transfer and feature fusion, which further boosts the performance. *Finally*, data augmentation for both modalities can be directly synchronized in the voxel space without the complex aligning process [12, 7].

The overall framework, called UVTR, can be easily instantiated and improved with various image- or voxel-based backbones for single- and multi-modality 3D object detection. Extensive empirical studies are conducted in Section 4 to reveal the effect of each component. The proposed UVTR attains leading performance in various settings. For detection, it achieves **69.7%**, **55.1%**, and **71.1%** NDS on nuScenes *test* set with point clouds, images, and multi-modality inputs, respectively. Given naive association strategy, UVTR also achieves strong tracking results with **67.0%**, **51.9%**, and **70.1%** AMOTA on LiDAR-based, camera-based, and multi-modality setting, respectively.

## 2 Related Work

**LiDAR-based 3D Detection.** With point clouds captured from LiDAR, traditional methods process the irregular input and generate 3D boxes with different representations, *e.g.,* point, voxel, and range view. Point-based detectors usually aggregate features from raw point clouds with set abstraction [13] and then predict box proposals [14, 15, 16]. For voxel-based methods, point clouds are transformed into regular grids and processed with 3D sparse convolutions [17, 18] or 2D convolutions [19, 20, 21] directly. Final predictions are usually generated on top of the bird-eye view (BEV) space with the flatted height axis [22, 23, 24]. There are also studies [3, 4] that project point clouds to range view and process them like images. However, due to the collapsed 3D geometry in range-view images,

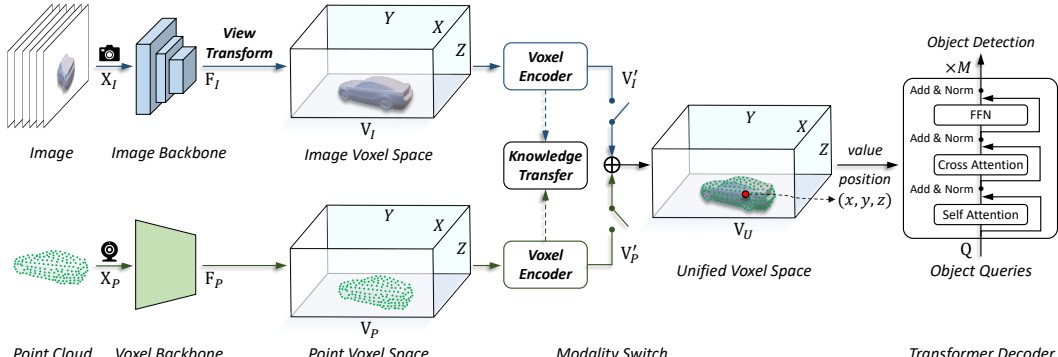

Figure 2: The framework of UVTR with multi-modality input. Given single- or multi-frame images and point clouds, we first process them in individual backbone and convert to modality-specific space $\mathbf{V}_I$ and $\mathbf{V}_P$, where view transform is utilized for that of image. In voxel encoder, features are spatially interacted, and knowledge transfer is easily supported during training. Single- or multi-modality features are selected via modality switch according to different settings. Finally, transformer decoder is utilized for prediction by sampling features from the unified space $\mathbf{V}_U$ with learnable positions.

the relationship in point clouds cannot be fully explored. In this work, we follow the voxel-based pipeline but keep the fine-grained voxel space without height compression, as shown in Figure 2.

**Camera-based 3D Detection.** Camera-based methods perform 3D detection on single- or multi-view images. With monocular image, previous approaches try to predict 3D boxes based on image features directly [25, 26, 27] or utilize the middle representation [1, 2, 5]. For multi-view input, image features are usually optimized in the constructed 3D geometry volume [28, 29]. Most recently, multi-view features are projected and merged in the frustum feature space with the aid of predicted depth [6]. Following the LiDAR-based paradigm, the frustum feature is collapsed to the BEV space, as briefly introduced in Figure 1c. However, the accuracy of the predicted depth map is much inferior to that of LiDAR, which brings semantic ambiguity to BEV space. Other recent studies try to capture geometry clues from multi-view images in an implicit manner [8, 10], which losses the chance for direct spatial interactions. In this paper, we represent image features in an explicit voxel space to alleviate the semantic ambiguity and facilitate further feature interactions, as depicted in Figure 1d.

**Cross-modality Interaction.** With input data from various sensors, cross-modality interaction is conducted to benefit from different inputs, *e.g.,* modality fusion and knowledge transfer. For modality fusion, the model takes data from different sensors and conducts fusion at point- and instance-level. Specifically, point-level fusion [30, 31, 32, 7] combines features from different modalities at the early stage of the network, which enables sufficient interaction. And instance-level fusion [33, 34, 35] is usually applied at the later stage to combine object-level features. Cross-modality knowledge transfer aims to distill specific knowledge [36] across modalities in the training phase. Compared with cross-modality fusion, knowledge transfer is seldom studied for 3D object detection. A prior work is LIGA-Stereo [37] that transfers geometry-aware representations from LiDAR to stereo images via distillation. Different from [37], UVTR represents each modality in a unified manner and supports cross-modality fusion and knowledge transfer simultaneously, which further enables distillation from multi-modality or consecutive frames to the single input.

## 3 UVTR Framework

The overall framework of UVTR is relatively simple: modality-specific space is constructed to unify the representation of inputs; cross-modality interaction is designed for feature learning across spaces; and transformer decoder is introduced for object-level interaction and final prediction.

### 3.1 Modality-specific Space

Given images $\mathbf{X}_I$ captured from cameras and point cloud $\mathbf{X}_P$ from LiDAR, different branches are utilized to respectively generate and enhance voxel space for each modality, as presented in Figure 2.

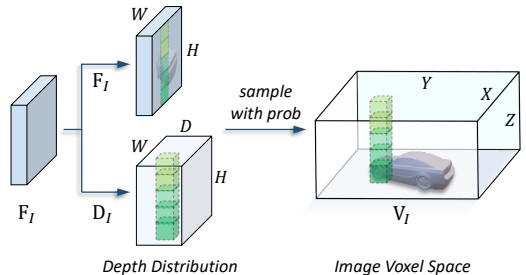
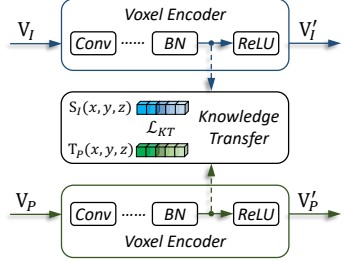

Figure 3: Details in the view transform.    Figure 4: Details in the knowledge transfer.

**Image Voxel Space.** For image voxel space, a shared backbone is adopted to extract features from multi-view or multi-frame images. In this process, FPN [38] is utilized for multi-scale context aggregation that is summed to formulate the feature $\mathbf{F}_I \in \mathbb{R}^{H \times W \times C}$, where $H$ and $W$ vary with FPN stages. To construct the voxel feature for images, we then transform the image feature of each view to the predefined space with the designed view transform in Figure 3. Motivated by [39, 5], we first generate the depth distribution $\mathbf{D}_I \in \mathbb{R}^{D \times H \times W}$ of each image with a single convolution as

$$\mathbf{D}_I(u, v) = \text{Softmax}(\text{Conv}(\mathbf{F}_I)(u, v)). \tag{1}$$

Here, $(u, v)$ indicates coordinate in the image plane, and $D$ is set to 64 to represent the perception limit 64*m*. It is noted that $\mathbf{D}_I$ is predicted without supervision. With the predicted $\mathbf{D}_I$ in $D$ depth bins, we can easily get the depth distribution of each pixel in $\mathbf{F}_I$. Let $(x, y, z)$ indicates a sampling point that is generated at the center of each bin from the voxel space $\mathbf{V}_I$. The point $(u, v, d)$ in the image plane is calculated from $(x, y, z)$ with the calibration matrix $\mathbf{P}$, where $d$ denotes the reference depth along axis $D$ of $\mathbf{D}_I$. Thus, the corresponding feature in voxel space $\mathbf{V}_I$ is easily captured by

$$\mathbf{V}_I(x, y, z) = \mathbf{D}_I(u, v, d) \times \mathbf{F}_I(u, v), \tag{2}$$

where $\mathbf{D}_I(u, v, d)$ represents the occupancy probability of feature $\mathbf{F}_I(u, v)$ in voxel $(x, y, z)$. For the multi-frame setting with $n$ sweeps, we use the shared network for all of them and formulate $n$ voxel spaces in total. In this process, each calibration matrix $\mathbf{P}$ is aligned to the ego vehicle in the initial frame. To gather temporal cues in each voxel space, relative time offsets from the initial frame are attached along the channel axis and merged using a single convolution. Then, $n$ voxel spaces are concatenated together, and the space-level fusion is conducted with a convolutional layer. In this way, features along the temporal dimension are integrated into a unified space $\mathbf{V}_I$, which is proved to bring significant gain in Table 3. Different from methods [5, 6] for BEV space, we preserve the 3D voxel space without collapsing in $Z$ axis to avoid the aforementioned semantic ambiguity and enable further interactions. The effectiveness of the 3D voxel space is empirically studied in Table 1.

**Point Voxel Space.** With the accurate position, we naturally split point cloud $\mathbf{X}_P$ into several regular voxels. Then, the voxel backbone in Figure 2 is utilized to process input voxels with sparse convolution [17]. To enhance multi-scale features in the generated voxel space, parallel heads with various strides are designed to extract feature $\mathbf{F}_P$ from the output. In particular, several 2D convolutions are applied in each head to aggregate the spatial cues at each height. Then, multi-scale features are upsampled to a same resolution and summed together to formulate the voxel space $\mathbf{V}_P \in \mathbb{R}^{X \times Y \times Z \times C}$. For multi-frame setting with $n$ sweeps, we follow previous work [24] and attach all point clouds together with relative time offsets to formulate the input $\mathbf{X}_P$.

Due to the accurate position of point cloud, the semantic ambiguity in $Z$ axis is much reduced compared with that of images. But we still preserve the 3D space $\mathbf{V}_P$ without height compression for convenient cross-modality interaction in Section 3.2 and fine-grained object interaction in Section 3.3. This is also proved to bring superior experimental results in Table 1.

**Voxel Encoder.** In the above-generated space $\mathbf{V}_I$, features of adjacent voxels projected from different views have no connection with each other. To solve this issue and facilitate local feature interaction, the voxel encoder is proposed in each voxel space, as presented in Figure 2. Specifically, we keep the simplicity of UVTR, and only three basic convolutional blocks are applied in each voxel encoder of Figure 4. In this process, features in each space $\mathbf{V}_I$ or $\mathbf{V}_P$ are aggregated in both coplanar and vertical dimensions. The spatial interaction in voxel space establishes connections among adjacent features, which is proved to be essential in Table 2, especially for $\mathbf{V}_I$.

## 3.2 Cross-modality Interaction

With the unified representation in space $\mathbf{V}_I$ and $\mathbf{V}_P$, interactions across modalities can be easily conducted. Given the prior that LiDAR is advanced in localization and cameras provide context for classification, the cross-modality interaction is proposed from two separate aspects, *i.e.*, transferring geometry-aware knowledge to images in a single-modality setting and fusing context-aware features with point clouds in a multi-modality setting. In particular, *knowledge transfer* aims to optimize the features of the student with guidance from the teacher in the single-modality setting. Meanwhile, *modality fusion* is designed to better utilize all modalities in both training and inference stages.

**Knowledge Transfer.** Considering *single modality* input in the inference stage, knowledge transfer is first designed to optimize features of the student with guidance from the teacher during training, which is important in an environment that lacks multi-modality data. Due to inherent properties, the geometry structure contained in images can be further exploited with the aid of point clouds, while the rich context in images can hardly be transferred to sparse point clouds. Therefore, we mainly focus on transferring knowledge from the geometry-rich modality to the poor one in this work. Benefiting from unified feature spaces, the cross-modality transfer can be easily supported, as illustrated in Figure 4. In particular, we take features before the last ReLU layer in the voxel encoder of $\mathbf{V}_P$ as the geometry-rich teacher, marked as $\mathbf{T}_P$. Meanwhile, the feature in the same position of $\mathbf{V}_I$ is taken as the geometry-poor student, denoted as $\mathbf{S}_I$. If we take one object query position $(x, y, z)$ from Section 3.3, the feature distance for knowledge transfer is formulated as

$$d_{KT} = PL_2(\mathbf{T}_P(x, y, z), \mathbf{S}_I(x, y, z)), \tag{3}$$

where $PL_2$ represents the partial $L_2$ distance [40]. Without bells-and-whistles, the optimization objective for knowledge transfer is averaged from $N$ object queries of transformer decoder in Section 3.3, namely $\mathcal{L}_{KT} = \frac{1}{N} \sum_i (d_{KT})$. It should be noted that the whole network is optimized in an end-to-end manner, with no need for extra procedures. Given the object position in each query, we can directly minimize the object-level distance with no need to exclude background features like [37]. In a similar pipeline, the knowledge transfer is further extended to support more input streams, like multi-frame images. The proposed cross-modality knowledge transfer is flexible with input modalities and brings consistent gains over various baselines in Tables 5 and 7.

**Modality Fusion.** Different from the knowledge transfer, modality fusion aims to better utilize *all modalities* in both training and inference stages, which utilizes the complementary knowledge of point cloud and images to improve the performance and robustness. Thanks to the unified representation of each modality, feature fusion can be naturally applied. To be specific, given the processed feature space $\mathbf{V}_I'$ and $\mathbf{V}_P'$, we first select candidate modality for final prediction via modality switch, as depicted in Figure 2. That means we support single- or multi-modality input for prediction according to different settings. If both modalities are taken, $\mathbf{V}_I'$ and $\mathbf{V}_P'$ are added together to formulate the unified voxel space $\mathbf{V}_U \in \mathbb{R}^{X \times Y \times Z \times C}$. In this way, both modalities are well expressed in a unified manner, which can be further fused with a single convolution. The space $\mathbf{V}_U$ unifies modalities with the explicit representation, which provides an expressive space for object interactions in Section 3.3.

## 3.3 Transformer Decoder

To obtain accurate and robust predictions, the transformer decoder is utilized for further object-level interaction in the unified voxel space $\mathbf{V}_U$. We draw inspirations from deformable DETR [11] and apply reference positions to efficiently sample representative features, regardless of the spatial size of 3D voxel spaces. In particular, we first initialize $N$ object queries $\mathbf{Q} \in \mathbb{R}^{N \times C}$ and generate $N$ reference points from object query embedding. Then, object queries are interacted with each other in the self-attention module and summed via skip-connection, as shown in Figure 2. Let $q$ represents a specific query in $\mathbf{Q}$ with corresponding reference point $p = (x, y, z)$. The process of the cross-attention module in Figure 2 is modeled as

$$\text{CrossAttn}(q, \mathbf{V}_U(p)) = \text{DeformAttn}(q, p, \mathbf{V}_U), \tag{4}$$

where $\mathbf{V}_U(p)$ denotes the sampled feature at $(x, y, z)$ of $\mathbf{V}_U$, and DeformAttn indicates the deformable attention in [11]. With the feed-forward network and normalization, each object query can easily interact with unified features from $\mathbf{V}_U$ inside each block. There are total $M$ blocks in the transformer decoder, where $M$ is respectively set to 3 and 6 for LiDAR-based and camera-based settings. Finally, a shared MLP head is utilized for prediction according to the output of each block. And iterative box refinement [11, 8] is applied to refine 3D bounding boxes based on the predictions.

### 3.4 Optimization Objectives

Following a general paradigm in recent transformers [41, 11], Hungarian algorithm [42] is adopted for one-to-one target assignment in the training phase. Thus, a set-to-set loss $\mathcal{L}_{Det}$ is computed to optimize detection results, including box regression loss and classification loss. If knowledge transfer is applied, the loss $\mathcal{L}_{KT}$ is contained to reduce cross-modality feature distance with a weight 0.01.

## 4 Experiments

In this section, we first introduce our detailed experimental setup. Then, analyses of each component are conducted on different modalities. Comparisons with several leading benchmarks on the nuScenes [43] dataset are presented in the end. More results are attached in supplementary material.

### 4.1 Experimental Setup

**Dataset.** nuScenes [43] dataset is a large-scale benchmark for autonomous driving, which is widely adopted for single- or multi-modality 3D object detection. It contains 700, 150, 150 scenes in the *train*, *val*, and *test* set, respectively. We use the synced data with 10 object categories that are captured from a 32-beam LiDAR at 20Hz and six cameras in a 360-degree field of view at 12Hz. Only annotations of keyframes are given at 2Hz. Here, ablation studies are optimized on a mini 1/4 *train* split by default, and final models are optimized on the whole *train* set.

**Implementation Details.** In this work, we conduct experiments on different modalities with 900 object queries **Q**. Constructed voxel spaces $\mathbf{V}_I$, $\mathbf{V}_P$, and $\mathbf{V}_U$ share the same shape $128 \times 128 \times Z$, where $Z$ indicates the height of voxel space and is further investigated in Table 1. The channel number $C$ in voxel spaces and transformer decoder is set to 256. And the amount of block $M$ in the decoder is set to 3, 6, and 6 for LiDAR-based, camera-based, and fusion settings, respectively. In particular, for the LiDAR-based setting, only the branch with voxel space $\mathbf{V}_P$ is kept. With grid size $0.1m$, the input point clouds are filtered in range $[-51.2m, 51.2m]$ for $X$ and $Y$ axis with $[-5.0m, 3.0m]$ for $Z$ axis. While for grid size $0.075m$, the range in $X$ and $Y$ axis is modified to $[-54.0m, 54.0m]$. The framework is trained with AdamW optimizer with an initial learning rate $2e^{-5}$ for 20 epochs. For a camera-based setting, the network is optimized with an initial learning rate $1e^{-4}$ for 24 epochs. As for fusion, we initialize two modality-specific branches with corresponding pretrained models and optimize the model with an initial learning rate $4e^{-5}$ for 20 epochs. The whole framework is trained in an end-to-end manner with different modalities. More details are given in supplementary material.

### 4.2 Component-wise Analysis

In *this subsection*, we use a randomly sampled 1/4 split of nuScenes *train* set for efficient validation.

**Effect of Height in Voxel Space.** As elaborated in Section 3.1, the height axis $Z$ plays a vital role in voxel space, especially for camera-based $\mathbf{V}_I$. In Table 1, we validate this with different heights on both modalities. Compared with the BEV space with height 1, the increase in height contributes significantly for camera-based $\mathbf{V}_I$, which respectively improves 3.1% and 4.2% NDS with height 5 and 11. Consistent with our analysis, the gain brought by increase of height contributes less to that of LiDAR because of the accurate position, which is up to 1% NDS and 1.9% mAP.

**Operations in Voxel Encoder.** The voxel encoder in Section 3.1 aims to facilitate spatial feature interactions that are essential, especially in the camera-based manner. As presented in Table 2, camera-based network cannot converge if given no spatial interaction. For LiDAR-based network, it performs satisfactorily without the voxel encoder, which can be attributed to the established relations at the early stage. Overall, for both of them, 3D spatial interaction still plays a vital role and improves 2.6% and 0.6% NDS over 2D convolution for the camera- and LiDAR-based methods, respectively.

**Effect of Multi-frame Input.** To further release the potential of the designed paradigm, we input multi-frame sweeps and represent them in each voxel space, as shown in Table 3. With more input frames, networks for both modalities achieve consistent gains. The performance gap reaches 5% and 18.1% NDS for the camera- and LiDAR-based manner with 5 and 10 sweeps, respectively.

**Networks for Voxel Space.** In Table 4, we exploit the network for voxel space generation. It is clear that the deeper network with larger voxel space contributes more to the final result. With ResNet-101

Table 1: Effect of different heights $Z$ in voxel space on nuScenes *val* set.

| modality | height | NDS(%) | mAP(%) |
|----------|--------|--------|--------|
| Camera | 1 | 31.4 | 24.9 |
| | 5 | 34.5 | 27.0 |
| | 11 | **35.6** | **28.7** |
| LiDAR | 1 | 62.8 | 54.4 |
| | 5 | 63.8 | 55.5 |
| | 11 | **63.8** | **56.3** |

Table 2: Effect of different operations in voxel encoder on nuScenes *val* set.

| modality | type | NDS(%) | mAP(%) |
|----------|------|--------|--------|
| Camera | None | 12.0 | 2.5 |
| | Conv2D | 31.9 | 24.8 |
| | Conv3D | **34.5** | **27.0** |
| LiDAR | None | 63.1 | 54.3 |
| | Conv2D | 63.2 | 54.6 |
| | Conv3D | **63.8** | **55.5** |

Table 3: Effect of different number of frames on nuScenes *val* set. *sweep* denotes the sweep number of multi-frame input.

| modality | sweep | NDS(%) | mAP(%) |
|----------|-------|--------|--------|
| Camera | 1 | 34.5 | 27.0 |
| | 3 | 38.0 | 28.7 |
| | 5 | **39.5** | **29.4** |
| LiDAR | 1 | 45.7 | 42.8 |
| | 10 | **63.8** | **55.5** |

Table 4: Effect of different models for voxel space construction on nuScenes *val* set. H and V denote space height and grid size.

| modality | voxel net | NDS(%) | mAP(%) |
|----------|-----------|--------|--------|
| Camera | R50-H5 | 34.5 | 27.0 |
| | R50-H11 | 35.6 | 28.7 |
| | R101-H11 | **39.4** | **32.0** |
| LiDAR | V0.1 | 63.8 | 55.5 |
| | V0.075 | **64.3** | **56.3** |

Table 5: Effect of different knowledge transfer settings on nuScenes *val* set. CS denotes multi-frame camera sweeps.

| student | teacher | NDS(%) | mAP(%) |
|---------|---------|--------|--------|
| Camera | – | 34.5 | 27.0 |
| | CS | 36.3 | 28.1 |
| | LiDAR | 36.4 | 28.2 |
| | Multi-mod | **37.1** | **28.8** |
| LiDAR | – | 63.8 | 55.5 |
| | Multi-mod | **64.4** | **56.1** |

Table 6: Effect of different network settings for cross-modality fusion on nuScenes *val* set. V denotes the split voxel grid size.

| camera | lidar | NDS(%) | mAP(%) |
|--------|-------|--------|--------|
| R50 | – | 34.5 | 27.0 |
| – | V0.1 | 63.8 | 55.5 |
| R50 | V0.1 | 65.1 | 59.0 |
| | V0.075 | **65.6** | **60.1** |
| R101 | V0.1 | 65.4 | 59.4 |
| | V0.075 | **66.3** | **61.0** |

and height 11 for space $\mathbf{V}_I$, the camera-based method attains nearly 5% gains over the baseline in both NDS and mAP. For LiDAR-based manner, the model performs slightly better with finer grid size that brings higher resolution to voxel space. It means the image-based voxel space requires strong backbones to extract expressive features, while the LiDAR-based one is less dependent on that.

**Knowledge Transfer.** Benefiting from the unified representation, knowledge can be easily transferred across modalities. In Table 5, we compare combinations with different students and teachers. In particular, the camera-based student captures geometry-aware cues from camera sweeps or the LiDAR-based teacher, which brings up to 1.9% NDS gain. If coupled with context features in the multi-modality setting, the gap is enlarged to 2.6% NDS and 1.8% mAP. For the LiDAR-based student, the increase brought by knowledge transfer saturated with 0.6% NDS. This can be attributed to that rich context in images can not be well expressed only with sparse points during inference. Therefore, images are required as input if rich context is supplemented, like the following fusion part.

**Cross-modality Fusion.** In Table 6, we validate capability of UVTR with cross-modality fusion. As presented in the table, feature fusion brings significant gains over a single modality. And the LiDAR-based representation dominates the final results, while the camera-based one provides supplementary context. It is reasonable because point clouds are more accurate in position and more representative in geometry expression. Cameras still provide sufficient context for better classification, which yields up to 1.6% NDS and 3.9% mAP gain compared with the LiDAR-based baseline. With finer voxel grids, the performance gap is enlarged to 2.5% NDS and 5.5% mAP.

Table 7: Comparisons of different methods with a single model on the nuScenes *val* set. We compare with classic methods on different modalities *without* test-time augmentation. [†] denotes the implementation from MMDetection3D [44]. L, C, CS, and M indicate the LiDAR, Camera, Camera Sweep, and Multi-modality input, respectively. L2 represents knowledge transfer from LiDAR.

| Method | Backbone | **NDS**(%) | **mAP**(%) | mATE↓ | mASE↓ | mAOE↓ | mAVE↓ | mAAE↓ |
|---|---|---|---|---|---|---|---|---|
| | | | *LiDAR-based* | | | | | |
| CenterPoint[†] [24] | V0.1 | 64.9 | 56.6 | 0.291 | 0.252 | 0.324 | 0.284 | 0.189 |
| UVTR-L | V0.1 | 66.4 | 59.3 | 0.345 | 0.259 | 0.313 | 0.218 | 0.185 |
| UVTR-L | V0.075 | **67.7** | **60.9** | 0.334 | 0.257 | 0.300 | 0.204 | 0.182 |
| | | | *Camera-based* | | | | | |
| DETR3D [8] | R101 | 42.5 | 34.6 | 0.773 | 0.268 | 0.383 | 0.842 | 0.216 |
| UVTR-C | R50 | 41.9 | 33.3 | 0.793 | 0.276 | 0.454 | 0.760 | 0.196 |
| UVTR-C | R101 | 44.1 | 36.2 | 0.758 | 0.272 | 0.410 | 0.758 | 0.203 |
| UVTR-CS | R50 | 47.2 | 36.2 | 0.756 | 0.276 | 0.399 | 0.467 | 0.189 |
| UVTR-CS | R101 | 48.3 | 37.9 | 0.731 | 0.267 | 0.350 | 0.510 | 0.200 |
| UVTR-L2C | R101 | 45.0 | 37.2 | 0.735 | 0.269 | 0.397 | 0.761 | 0.193 |
| UVTR-L2CS | R101 | **48.8** | **39.2** | 0.720 | 0.268 | 0.354 | 0.534 | 0.206 |
| | | | *LiDAR+Camera* | | | | | |
| FUTR3D [9] | V0.075-R101 | 68.3 | 64.5 | - | - | - | - | - |
| UVTR-M | V0.075-R101 | **70.2** | **65.4** | 0.332 | 0.258 | 0.268 | 0.212 | 0.177 |

Table 8: Comparisons of different distances, weather, and lighting conditions on nuScenes *val* set.

| Method | Modality | Distance: NDS(%) | | | Weather: NDS(%) | | Lighting: NDS(%) | |
|---|---|---|---|---|---|---|---|---|
| | | *<20m* | *20-30m* | *>30m* | *Sunny* | *Rainy* | *Day* | *Night* |
| CenterPoint [24] | LiDAR | 74.1 | 62.1 | 34.6 | 64.6 | 64.4 | 65.1 | 40.1 |
| UVTR-L | LiDAR | 75.9 | 64.9 | 37.3 | 67.4 | 67.9 | 67.8 | 41.4 |
| UVTR-C | Camera | 52.8 | 39.7 | 20.4 | 43.1 | 48.3 | 44.5 | 23.5 |
| UVTR-M | Multi-mod | **77.2** | **68.2** | **38.9** | **69.7** | **72.0** | **70.3** | **42.6** |

## 4.3 Main Results

In this section, we first report results with various modalities that are optimized on the whole *train* set. Then, we give analyses of the framework robustness, including camera view drop and translational noise. Comparisons with leading methods on the nuScenes *test* set are presented in the end.

**Results with Different Modalities.** In Table 7, we carry out experiments with different modalities on the nuScenes *val* set. Compared with classic methods, UVTR achieves significant improvement. In particular, for LiDAR-based method, it attains 1.5% NDS and 2.7% mAP gain over CenterPoint [24]. And a finer resolution contributes better results with 67.7% NDS. For camera-based manner, UVTR-C performs better in single frame setting with 1.6% NDS gain over DETR3D [8]. If applied knowledge transfer in UVTR-L2C, the gap is enlarged to 2.5% NDS. The performance improves with more frames and attains up to 48.8% NDS. For multi-modality input, UVTR-M achieves 4.5% mAP gain over UVTR-L and outperforms the contemporary FUTR3D [9] with 1.9% NDS in a same setting.

**Results in Different Conditions.** In Table 8, we report the performance with different distances, weather conditions, and light situations. (1) Distance: For LiDAR-based approaches, the proposed UVTR-L achieves better performance in all situations compared with CenterPoint [24]. Equipped with both LiDAR and camera inputs in UVTR-M, the framework attains significant gains, especially in a relatively far distance (3.3% NDS gain in 20-30*m* ). If the object is too far (>30*m*), the performance gain decreases to 1.6% NDS, but still much better than CenterPoint and UVTR-L. (2) Weather condition: It is clear that the proposed UVTR-L achieves significant gain compared with CenterPoint in both conditions. And additional camera input brings much better results, especially in rainy weather (4.1% NDS gain). (3) Light situation: Compared with that in the daylight situation, both LiDAR-based and camera-based approaches perform inferior in the dark night. Compared with

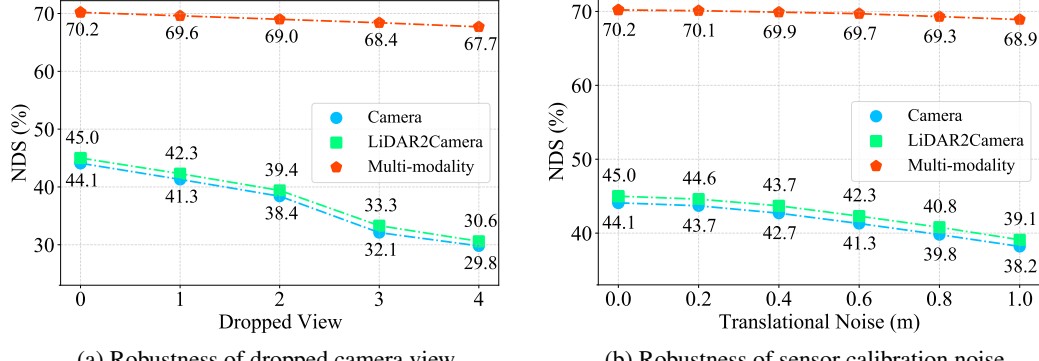

(a) Robustness of dropped camera view      (b) Robustness of sensor calibration noise

Figure 5: We validate the robustness of UVTR by adding two typical errors during inference. For dropped view in 5a, we randomly drop a fixed number of camera views to simulate the camera failure. For sensor noise in 5b, we randomly add translational noises in LiDAR to camera calibration matrix.

Table 9: Comparisons of leading methods with a single model on the nuScenes *test* set. L, C, CS, and M indicate the LiDAR, Camera, Camera Sweep, and Multi-modality input, respectively. L2 represents knowledge transfer from LiDAR. Flipping augmentation is adopted for LiDAR. It should be noted that the performance of UVTR-L2CS3 can be further improved with more than 3 sweeps.

| Method | Backbone | NDS(%) | mAP(%) | mATE↓ | mASE↓ | mAOE↓ | mAVE↓ | mAAE↓ |
|---|---|---|---|---|---|---|---|---|
| *LiDAR-based* | | | | | | | | |
| 3DSSD [45] | Point-based | 56.4 | 42.6 | - | - | - | - | - |
| CenterPoint [24] | V0.075 | 65.5 | 58.0 | - | - | - | - | - |
| HotSpotNet [46] | V0.1 | 66.0 | 59.3 | 0.274 | 0.239 | 0.384 | 0.333 | 0.133 |
| AFDetV2 [47] | V0.075 | 68.5 | 62.4 | 0.257 | 0.234 | 0.341 | 0.299 | 0.137 |
| UVTR-L | V0.075 | **69.7** | **63.9** | 0.302 | 0.246 | 0.350 | 0.207 | 0.123 |
| *Camera-based* | | | | | | | | |
| FCOS3D [27] | R101 | 42.8 | 35.8 | 0.690 | 0.249 | 0.452 | 1.434 | 0.124 |
| DD3D [48] | V2-99 | 47.7 | 41.8 | 0.572 | 0.249 | 0.368 | 1.014 | 0.124 |
| DETR3D [8] | V2-99 | 47.9 | 41.2 | 0.641 | 0.255 | 0.394 | 0.845 | 0.133 |
| BEVDet [6] | V2-99 | 48.8 | 42.4 | 0.524 | 0.242 | 0.373 | 0.950 | 0.148 |
| PETR [10] | V2-99 | 50.4 | 44.1 | 0.593 | 0.249 | 0.383 | 0.808 | 0.132 |
| UVTR-L2C | V2-99 | 52.2 | 45.2 | 0.612 | 0.256 | 0.385 | 0.664 | 0.125 |
| UVTR-L2CS3 | V2-99 | **55.1** | **47.2** | 0.577 | 0.253 | 0.391 | 0.508 | 0.123 |
| *LiDAR+Camera* | | | | | | | | |
| FusionPainting [49] | V0.075-R50 | 70.4 | 66.3 | - | - | - | - | - |
| MVP [32] | V0.075-DLA34 | 70.5 | 66.4 | - | - | - | - | - |
| PointAugmenting [50] | V0.075-DLA34 | 71.0 | 66.8 | - | - | - | - | - |
| UVTR-M | V0.075-R101 | **71.1** | **67.1** | 0.306 | 0.245 | 0.351 | 0.225 | 0.124 |

CenterPoint, the proposed UVTR-L still performs better. And the camera inputs still bring significant gains in both situations, especially in a daylight environment (2.5% NDS gain).

**Robustness of the Framework.** To validate the robustness of UVTR, we simulate two typical sensor errors during inference in Figure 5. For loss of view, the multi-modality manner achieves well robustness in Figure 5a. Because LiDAR can still capture surrounding scenes if cameras are offline. As for the camera-based manner, the model losses inputs from dropped scenes, but the network still works well and outputs predictions within the field of view. For sensor jitter in Figure 5b, the model performs stable especially in the multi-modality setting because of the accurate perception from LiDAR. Meanwhile, knowledge transfer consistently improves performance in both settings.

**Comparison with Leading Methods.** In Table 9, we present comparisons with leading methods on the nuScenes *test* set. For LiDAR-based method, UVTR-L surpasses AFDetV2 [47] with 1.2% NDS and attains 69.7% NDS. For camera-based manner, with single frame, UVTR-L2C outperforms

Table 10: Comparisons of different leading tracking methods on nuScenes *test* set. * indicates the method at the leaderboard with no released publication.

| Method | Tracker | **AMOTA**(%) | AMOTP | Recall |
|---|---|---|---|---|
| *LiDAR-based* | | | | |
| CenterPoint [24] | Greedy | 63.8 | 0.555 | 0.675 |
| UVTR-L | Greedy | **67.0** | 0.656 | 0.703 |
| *Camera-based* | | | | |
| PolarDETR [51] | Transformer | 27.3 | 1.185 | 0.404 |
| BEVTrack* | Private | 34.1 | 1.107 | 0.463 |
| UVTR-L2CS3 | Greedy | **51.9** | 1.125 | 0.599 |
| *LiDAR+Camera* | | | | |
| EagerMOT [52] | Two-stage | 67.7 | 0.550 | 0.727 |
| AlphaTrack [53] | Position+Appearance | 69.3 | 0.585 | 0.723 |
| UVTR-M | Greedy | **70.1** | 0.686 | 0.750 |

PETR [10] with 1.8% NDS and 1.1% mAP. With 3 camera sweeps, UVTR-L2CS3 obtains significant gain and attains 55.1% NDS, which can be further improved with more frames. For the multi-modality setting, we directly adopt pretrained models from LiDAR- and camera-based manner with simple fine-tuning without bells-and-whistles. Compared with similar approaches without special module, UVTR-M achieves 71.1% NDS and 67.1% mAP, which is on par with leading approaches.

**Tracking Extension.** To better illustrate the capability and generality of the proposed UVTR, we further conduct experiments on the downstream tracking task. In particular, we follow the classic tracking-by-detection paradigm and apply the simple greedy tracker in CenterPoint. The only difference lies in that we adopt threshold 0.2 and NMS to remove low quality results. As presented in Table 10, the proposed UVTR achieves leading tracking performance with the greedy tracker in different settings. Specifically, in a camera-based setting, the proposed UVTR-L2CS3 surpasses previous SOTA at the leaderboard (BEVTrack) with **17.8**% AMOTA. It further proves the effectiveness and generality of the proposed cross-modality interaction in UVTR.

## 5  Discussion and Conclusion

We presented the UVTR, a conceptually simple yet effective framework for multi-modality 3D object detection. The key innovation lies in that it unifies the voxel-based representation for different modalities and facilitates multi-level interactions. In particular, it uniformly encodes inputs from different sensors in the modality-specific space to reduce the semantic ambiguity and enable spatial interaction. With the unified representation, the cross-modality interaction can be easily conducted for knowledge transfer and modality fusion. Moreover, object-level interactions in the unified space are further supported by the transformer decoder for accurate and robust detection. Experiments on the nuScenes dataset prove the effectiveness of UVTR, which attains consistent improvements over various benchmarks and surpasses previous methods with leading performance.

There still exist certain limitations in the current method. First, to construct voxel space for multi-view images, we need to process all of them in the shared image backbone, which brings computational cost, especially for multi-frame setting. In the future, we plan to explore a new manner for voxel space construction at the early stage of the network, like that of point clouds. Moreover, we construct the voxel space with a spatial resolution $128 \times 128$. We believe a higher resolution and more image frames could bring stronger voxel space with better results, which remains to be explored.

**Societal Impacts.** The proposed method focuses on 3D object detection that can be used in autonomous driving. Theoretically, a better 3D detector leads to safer autonomous vehicles. However, in the short term, the current technique could not solve all the corner cases and extreme situations. It may bring potential risks to the decision process in real-world autonomous systems.

**Acknowledgement.** This work is supported by Shenzhen Science and Technology Program KQTD20210811090149095.

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
