# Unifying Voxel-based Representation with Transformer for 3D Object Detection
## *Supplementary Material*

**Yanwei Li**[1]    **Yilun Chen**[1]    **Xiaojuan Qi**[2]    **Zeming Li**[3]    **Jian Sun**[3]    **Jiaya Jia**[1,4]

The Chinese University of Hong Kong[1]    The University of Hong Kong[2]
MEGVII Technology[3]    SmartMore[4]

## A    Experimental Details

In this section, we provide more technical details of the proposed UVTR. Because of the inherent properties, LiDAR- and camera-based methods usually adopt different pipelines for network training. For instance, GT-sampling and global augmentation are adopted for LiDAR-based approaches, while image-level augmentations are used for camera-based manners. To bridge the modality gap, we utilize the unified sampler and augmentation for network optimization.

**Unified Sampler.** Given point clouds of the captured scene, traditional LiDAR-based methods usually adopt GT-sampling [22] to supplement more samples from the whole database. However, due to the object overlapping in each view, this is seldom used for camera-based manners. In this work, we follow previous studies [50, 7] and use a unified approach for GT-sampling. In particular, we first generate point clouds and image crops of each sample from training scenes. For the multi-modality setting like UVTR-M in Tables 7 and 9, point clouds of sampled objects are attached to the original scene, and image crops are reorganized according to the actual depth and then pasted on the original image. For LiDAR-based settings, only point clouds are sampled for training, which is the same as previous work. The unified sampler is disabled at the last 2 epochs to fit the normal distribution.

**Unified Augmentation.** With the unified representation, global augmentations can be synchronized in the voxel space. Specifically, given the widely-adopted global scaling, rotation, and flipping for point clouds, we apply the same augmentations to the image voxel space $\mathbf{V}_I$. That means we first construct the voxel space $\mathbf{V}_I$ for images and then conduct space-level augmentations to stay the same with that of point cloud $\mathbf{V}_P$. In this manner, both modalities are well aligned in augmentation for cross-modality interactions and the following object-level interaction in the transformer decoder. For LiDAR-based and multi-modality settings, we adopt all the augmentations during training. For camera-based settings, only global scaling and rotation are applied to image voxel space $\mathbf{V}_I$.

**Training Schedule.** In Section 4.1 of the main paper, we give training details with different modalities. To be more specific, for the multi-modality setting, we finetune the framework for 20 epochs, which may not be fully optimized. Following previous work [50, 7], CBGS sampler is utilized for class balance optimization in the training process. Specifically, we initialize the camera- or LiDAR-based branch from the corresponding pretrained model and reduce the total training epoch to 10. In Table 11, we compare with previous methods on the nuScenes dataset. Obviously, the proposed UVTR can further be improved to 70.6% NDS and 65.9% mAP with CBGS on the nuScenes *val* set.

**Training Setting.** Due to different pipelines, we construct each training batch on 8 devices with 32, 8, and 16 input data for LiDAR-, camera-based, and multi-modality settings, respectively. For camera-based setting, we initialize our image backbone from the pretrained FCOS3D [27]. Most of our models are trained on NVIDIA V100 GPU. Part of memory-consuming models with multi-frame or multi-modality settings like UVTR-CS and UVTR-M are trained on NVIDIA A100 GPU. Here, we also provide the comparison of different distances for knowledge transfer in Table 12. The partial

Table 11: Comparisons of different methods with a single model on the nuScenes *val* set. We compare with classic methods on the multi-modality setting. M indicates the Multi-modality input.

| Method | Backbone | NDS(%) | mAP(%) | mATE↓ | mASE↓ | mAOE↓ | mAVE↓ | mAAE↓ |
|---|---|---|---|---|---|---|---|---|
| *LiDAR+Camera* | | | | | | | | |
| FUTR3D [9] | V0.075-R101 | 68.3 | 64.5 | - | - | - | - | - |
| UVTR-M | V0.075-R101 | 70.2 | 65.4 | 0.332 | 0.258 | 0.268 | 0.212 | 0.177 |
| UVTR-M-CBGS | V0.075-R101 | **70.6** | **65.9** | 0.320 | 0.256 | 0.262 | 0.219 | 0.176 |

Table 12: Comparisons between L2 and partial L2 distance [40] on the nuScenes *val* set. Models are optimized on 1/4 mini nuScenes *train* set. L2C represents knowledge transfer from LiDAR.

| Method | Backbone | Distance | NDS(%) | mAP(%) |
|---|---|---|---|---|
| UVTR-L2C | R50 | L2 | 36.0 | 28.0 |
| UVTR-L2C | R50 | Partial L2 | **36.4** | **28.2** |

Table 13: Model inference runtime on the nuScenes *val* set. We test all the models and report results on a single NVIDIA Tesla V100 GPU.

| Method | Backbone | NDS(%) | Latency(*ms*) | | | | FPS |
|---|---|---|---|---|---|---|---|
| | | | Backbone | ViewTrans | Encoder | Decoder | |
| UVTR-L | V0.1 | 66.4 | 71.5 | - | 17.1 | 18.4 | 9.3 |
| UVTR-C | R50 | 41.9 | 103.4 | 64.1 | 32.1 | 36.5 | 4.2 |
| UVTR-C | R101 | 44.1 | 194.1 | 64.7 | 32.3 | 36.1 | 3.1 |

Table 14: Comparisons with convolution-based head in the nuScenes *val* set. CPHead indicates the adopted convolution-based head in CenterPoint [24].

| Method | Backbone | Head | NDS(%) | mAP(%) |
|---|---|---|---|---|
| *LiDAR* | | | | |
| CenterPoint [24] | V0.1 | Convolution | 64.9 | 56.6 |
| UVTR-L-CPHead | V0.1 | Convolution | 65.4 | 58.1 |
| UVTR-L | V0.1 | Transformer | **66.4** | **59.3** |
| *Camera* | | | | |
| UVTR-C-CPHead | R101 | Convolution | 40.0 | 35.1 |
| UVTR-C | R101 | Transformer | **44.1** | **36.2** |

L2 distance brings a 0.4% NDS gain compared with the naive version. Therefore, we adopt partial L2 distance for knowledge transfer by default.

**Model Inference.** In the inference stage, we keep 300 top-scoring predictions within the range $[-10m, -10m]$ for $Z$ axis and $[-61.2m, -61.2m]$ for $X$ and $Y$ axis. We also provide the inference speed of the framework in Table 13. For the LiDAR-based setting, UVTR-L consumes cost mainly from the sparse convolution backbone. And the transformer decoder with 3 layers costs about 18*ms*. For the camera-based setting, the consumption is mainly from the image backbone and view transform process. The voxel encoder and transformer decoder with 6 layers also bring a noticeable cost. In this work, we focus more on the unified representation with good performance. And the framework can be further accelerated with several engineering skills. For example, we directly adopt the naive grid sampling in the view transform. It can be optimized a lot using a CUDA version operator.

**Decoder Head.** The transformer decoder is designed for object-level interaction and efficient object feature capture from the voxel space $\mathbf{V}_U$. Here, we compare it with the classic convolution-based CenterPoint [24] head in Table 14. In particular, to suit the CenterPoint head without bringing too much cost, we compress the axis $Z$ of the constructed unified voxel space $\mathbf{V}_U$ (after voxel encoder) in Figure 2 with a summation. As presented in Table 14, with the same head, the proposed UVTR

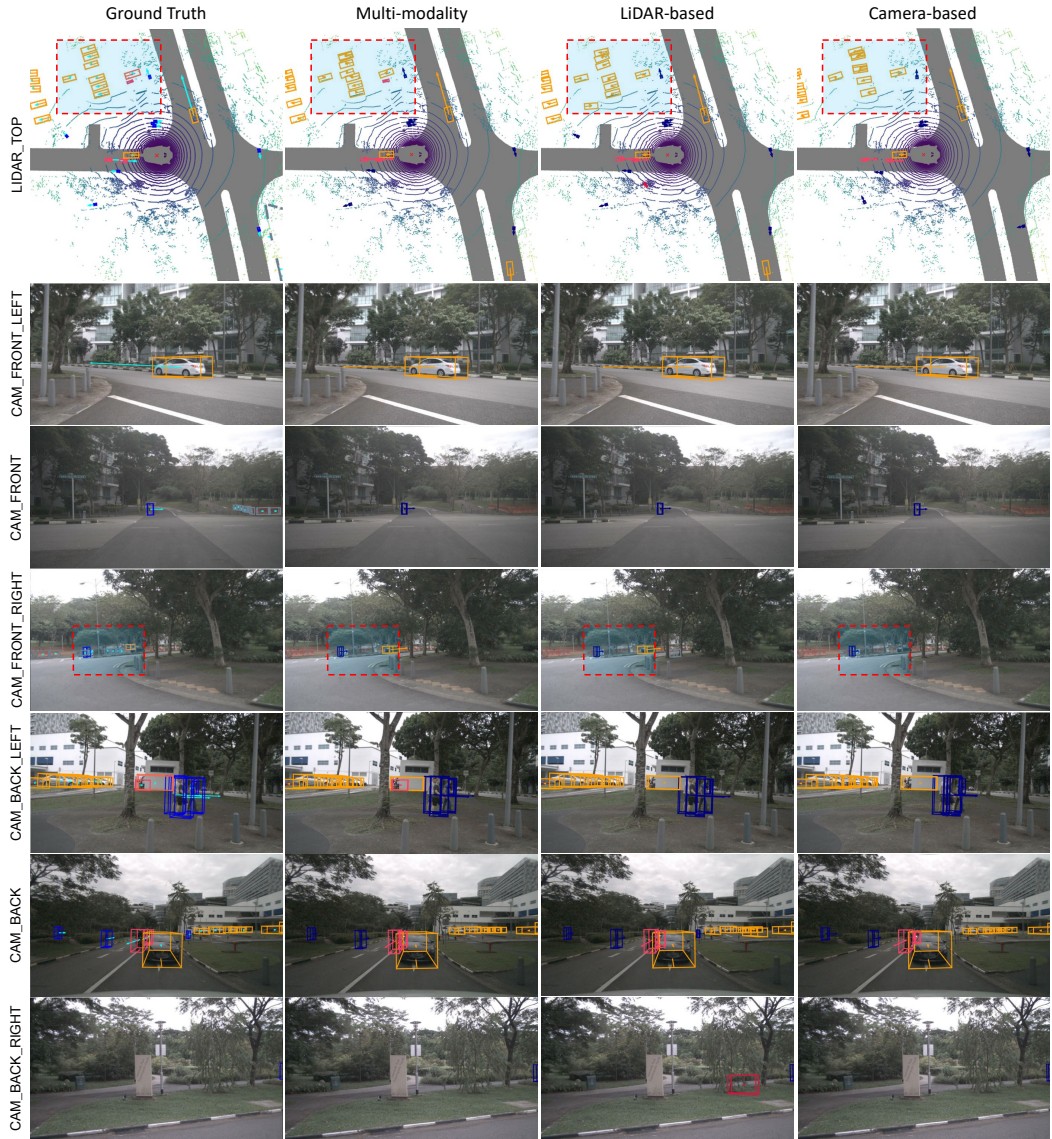

Figure 6: Visualization of UVTR predictions with different modalities on nuScenes *val* set. The important area that needs more attention is marked with dotted red box. Best viewed in color.

still surpasses CenterPoint with 0.5% NDS and 1.5% mAP. Compared with the convolution-based head, the designed transformer head achieves significant gains with 1.0% NDS and 4.1% NDS for LiDAR-based and camera-based settings, respectively. This proves the effectiveness of the designed transformer decoder in UVTR.

## B    Qualitative Analysis

In this section, we give visualizations of UVTR predictions on different modalities and different views, as presented in Figures 6 and 7. We draw the predicted results on LiDAR-based BEV views and each camera view for clear comparisons. It is clear that UVTR performs well on nuScenes [43] dataset, and most of the objects are detected in these scenes.

**Multi-modality Results.** The multi-modality results are given in the second column of Figures 6 and 7. Compared with the ground truth, the multi-modality detector gives accurate predictions of object location, category, speed, and orientation (arrows in each figure). Although the multi-

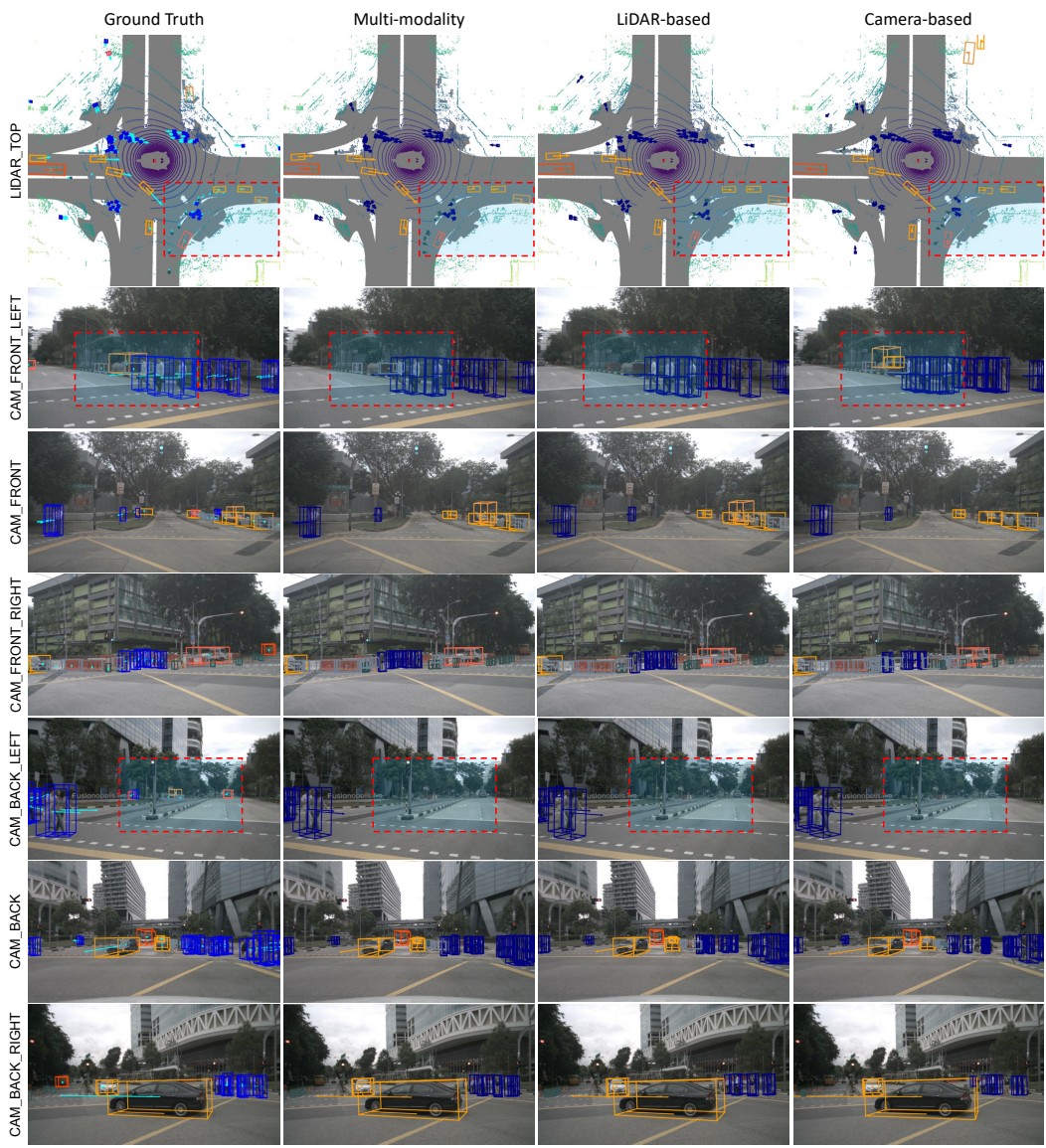

Figure 7: Extended visualization of UVTR predictions with different modalities on nuScenes *val* set. The important area that needs more attention is marked with dotted red box. Best viewed in color.

modality setting introduces advantages from both LiDAR and camera, there still exists missing objects, especially for far or small objects. It remains potential to be further explored for these cases.

**LiDAR-based Results.** For LiDAR-based detector, predictions are presented in the third column of Figures 6 and 7. Different from that of multi-modality results, the LiDAR-based method lacks sufficient context from images for accurate classification and thus brings the wrong detection. For example, without surrounding context, the *tree* in last row of Figure 6 is detected as *vehicle*.

**Camera-based Results.** For camera-based approach, we plot results in the last column of Figures 6 and 7. Because accurate positions are missing in images, locations of predicted boxes are not so accurate. But the camera-based manner provides more context cues for better recognition. As shown in the second row of Figure 7, *barriers* are well detected with the aid of images for camera-based and multi-modality methods, while this is not the case for LiDAR-based manner.