# OpenReview forum: "Unifying Voxel-based Representation with Transformer for 3D Object Detection"
_NeurIPS.cc/2022/Conference — NeurIPS 2022 Accept_

### Official Review · Reviewer_Bb8v · 2022-07-10

**Rating:** 4
**Confidence:** 4
**Soundness:** 2 fair
**Presentation:** 2 fair
**Contribution:** 2 fair

**Summary:**

In this paper, the authors proposed a 3D detection framework by projecting various sensor data, which are acquired at the same time, onto a voxel feature space. They first point out the limitation of conventional methods where input- or feature-level data are represented as 2D data with reduced dimensions (range image, predicted depth map, etc.). To overcome these types of issues, they introduce a unified voxel space without loss of spatial information. Additionally, the authors make the neural network more effective in extracting sensor fusion features by applying for knowledge transfer / Data augmentation without the complex aligning process.

**Questions:**

I think the authors' cross-modality interaction module should be validated in a more extreme environment. Rather than simply comparing whether features extracted from individual sensors are used, it is necessary to test results in an environment where one of the two sensors does not work at all. (For example, experiments on whether detection is possible only with camera data at a long distance where lidar sensor data does not exist, and experiments on cases where the camera does not work well at night or when passing through a tunnel entrance)

Introduction of deformable attention for cross attention: Unlike the image space, which contains various scales, the size of the object is determined in the actual 3D space. Therefore, it is not necessary to obtain information of various scales using deformable convolution. Please explain the reason for using deformable attention in unified voxel space. Additionally, can the authors attach a comparison with a network using local attention (basic conv.)?


**Limitations:**

The authors mentioned the shortcomings of their detector and the direction to be taken in the future. The disappointing point is that they restate known limitations, where voxel-based methods have a large increase in computation voxel-based methods have a large increase in the amount of computation compared to other representation methods (e.g., range view / bird’s eye view), in this field.

**Strengths And Weaknesses:**

(+) They improved the performance of the 3D detector by devising a sensor fusion method on the voxel representation that is richer than the bird-eye's view representation. Through sections 3 and 4 and the supplementary, the authors provide sufficient information to reconstruct and train the proposed network.

(-) The relationship between modality transfer learning techniques and cross-modality interaction feels awkward. As mentioned by the authors, Image and Lidar points are already complementary to each other, so just using the two features should improve performance. However, the knowledge transfer method seems to improve performance by making the features extracted from images similar to the features extracted from the lidar sensor data.

(-) The authors need to conduct performance evaluation in various environments to reveal the advantages of sensor fusion well.

---

> ### Author Response · Authors · 2022-08-02
> **Author response to Reviewer Bb8v (1/2)**
>
> Dear Reviewer *Bb8v*,
>
> Thank you for the valuable suggestions. We address your questions below.
>
> **Q1: The relationship between knowledge transfer and modality fusion feels awkward.**
>
> A1: Sorry for the misunderstanding. The knowledge transfer and modality fusion are actually separate parts of cross-modality interaction in Section 3.2. **(1)** Knowledge transfer: As declared in L148-L149  of the main paper, *knowledge transfer* aims to optimize the features of the student with guidance from the teacher in the *single-modality* setting. For example, in the camera-based setting, the point cloud is utilized *during training only* to provide geometry prior for the optimization guidance of image features. In this way, the network is able to achieve better performance *with images only* during inference. This could be important in an environment that lacks LiDAR data. **(2)** Modality fusion: As illustrated in L165-L166 of the main paper, *modality fusion* is designed to better utilize all modalities in both training and inference stages. In modality fusion, we take the complementary knowledge of point cloud and images to improve the performance as you mentioned. We will revise the description in Section 3.2 to make their relationship clear.
>
> **Q2: Evaluate the model in various and more extreme environments.**
>
> A2: Thanks for this suggestion. We follow your suggestion and report the performance of *different distances* in Table A-5, *different weather conditions* in Table A-6, and *different light situations* in Table A-7. Because it is hard to validate such specific scenes, like a tunnel entrance. Here we give analyses of similar environments with long distances, rainy weather, and dark night.
>
> **(1)** Distance: In Table A-5, we report performance with different modalities for input at various distances. For LiDAR-based approaches, the proposed UVTR-L achieves better performance in all situations compared with CenterPoint [24]. Equipped with both LiDAR and camera inputs in UVTR-M, the framework attains significant gains, especially in a relatively far distance (3.3% NDS gain in 20~30$m$). If the object is too far (>30$m$), the performance gain decreases to 1.6% NDS, but still much better than CenterPoint and UVTR-L.
>
> **(2)** Weather condition: In Table A-6, we conduct experiments on different weather conditions, i.e., sunny and rainy. It is clear that the proposed UVTR-L achieves significant gain compared with CenterPoint in both conditions. And additional camera input brings much better results, especially in rainy weather (4.1% NDS gain).
>
> **(3)** Light situation: We perform experiments on the mentioned night situation in Table A-7. Compared with that in the daylight situation, both LiDAR-based and camera-based approaches perform inferior in the dark night, as presented in Table A-7. Compared with CenterPoint, the proposed UVTR-L still performs better. And the camera inputs still bring significant gains in both situations, especially in a daylight environment (2.5% NDS gain). These tables will be summarized in the final revision.
>
> Table A-5: Comparisons among different distances in the nuScenes *val* set. We separate the dataset into three splits according to the object distance.
>
> | Method           | Modality | NDS(%) [<20$m$] | NDS(%) [20~30$m$] | NDS(%) [>30$m$] |
> | ---------------- | -------- | --------------- | ----------------- | --------------- |
> | CenterPoint [24] | LiDAR    | 74.1            | 62.1              | 34.6            |
> | **UVTR-L**       | LiDAR    | 75.9            | 64.9              | 37.3            |
> | **UVTR-C**       | Camera   | 52.8            | 39.7              | 20.4            |
> | **UVTR-M**       | Both     | **77.2** (+1.3) | **68.2** (+3.3)   | **38.9** (+1.6) |
>
> Table A-6: Comparisons between different weather conditions in the nuScenes *val* set. We separate the dataset into two splits according to the weather description.
>
> | Method           | Modality | Backbone    | NDS(%) [Sunny]  | NDS(%) [Rainy]  |
> | ---------------- | -------- | ----------- | --------------- | --------------- |
> | CenterPoint [24] | LiDAR    | V0.075      | 64.6            | 64.4            |
> | **UVTR-L**       | LiDAR    | V0.075      | 67.4            | 67.9            |
> | **UVTR-C**       | Camera   | R101        | 43.1            | 48.3            |
> | **UVTR-M**       | Both     | V0.075-R101 | **69.7** (+2.3) | **72.0** (+4.1) |

---

> ### Author Response · Authors · 2022-08-02
> **Author response to Reviewer Bb8v (2/2)**
>
> Table A-7: Comparisons between different light conditions in the nuScenes *val* set. We separate the dataset into two splits according to the light description.
>
> | Method           | Modality | Backbone    | NDS(%) [Day]    | NDS(%) [Night]  |
> | ---------------- | -------- | ----------- | --------------- | --------------- |
> | CenterPoint [24] | LiDAR    | V0.075      | 65.1            | 40.1            |
> | **UVTR-L**       | LiDAR    | V0.075      | 67.8            | 41.4            |
> | **UVTR-C**       | Camera   | R101        | 44.5            | 23.5            |
> | **UVTR-M**       | Both     | V0.075-R101 | **70.3** (+2.5) | **42.6** (+1.2) |
>
> **Q3: Why introduce deformable attention for cross attention?**
>
> A3: Sorry for the misunderstanding. The *deformable attention* in the decoder is introduced mainly for the model efficiency, regardless of scale variance (not the *deformable convolution*). As declared in A1 of the *response to all reviewers*, the deformable attention generates sampling point $(x,y,z)$ for each object query and performs object-level interaction in a sparse manner, regardless of the spatial size of feature maps. Moreover, the usage of deformable attention also brings faster convergence speed during training [11]. We will make this motivation clear in the revision.
>
> **Q4: Comparison with a network using local attention or basic conv.**
>
> A4: Thanks for this suggestion. We guess *"the network using basic conv"* means the convolution-based head. If we misunderstand it, please rectify us and we will try to provide more detailed results. In this way, we provide a classic convolutional head (CenterPoint head) for comparisons in A2 of the *response to all reviewers*. It is clear that the proposed transformer decoder achieves significant gains over the convolution-based head. Please refer to Table A-1 for more details. We will add this comparison in the revision.

---

> > ### Comment · Reviewer_Bb8v · 2022-08-09
> > **Responses to Authors**
> >
> > First, I would like to thank the authors for conducting experiments in various environments (distance, weather conditions, light situation) and basic convolution head. Also, the additional explanation of the attention part helped me understand this work better. In particular, concerns were resolved through additional experiments.
> >
> > But I still think the writing needs to be polished overall. So, assuming they rewrite what they mentioned, including section 3.2, I'll score it up as a "borderline accept".

---

> > > ### Author Response · Authors · 2022-08-09
> > > **Author response to Reviewer Bb8v**
> > >
> > > Dear Reviewer Bb8v,
> > >
> > > We sincerely thank your feedback and support. Definitely, we will polish the whole framework and also rewrite the confusing part following your suggestions. Moreover, to provide more details and make our solution clear, we will release the code and models for both detection and the following tracking part to the public.

---

> > > ### Author Response · Authors · 2022-08-10
> > > **Author response to Reviewer Bb8v-Revision of Section 3.2**
> > >
> > > Dear Reviewer Bb8v,
> > >
> > > We follow your suggestions and keep polishing the paper. Because the revision cannot be uploaded now, we attach the revision of Section 3.2 below. Hope it can address your remaining concern.
> > >
> > > **3.2 Cross-modality Interaction**
> > >
> > > With the unified representation in space $\mathbf{V}_I$ and $\mathbf{V}_P$, interactions across modalities can be easily conducted. Given the prior that LiDAR is advanced in localization and cameras provide context for classification, the cross-modality interaction is proposed from two separate aspects,  *i.e.*, transferring geometry-aware knowledge to images in a single-modality setting and fusing context-aware features with point clouds in a multi-modality setting. In particular, *knowledge transfer* aims to optimize the features of the student with guidance from the teacher in the single-modality setting. Meanwhile, *modality fusion* is designed to better utilize all modalities in both training and inference stages.
> > >
> > > **Knowledge Transfer.** Considering *single modality* input in the inference stage, knowledge transfer is first designed to optimize features of the student with guidance from the teacher during training, which is important in an environment that lacks multi-modality data. Due to inherent properties, the geometry structure contained in images can be further exploited with the aid of point clouds, while the rich context in images can hardly be transferred to sparse point clouds. Therefore, we mainly focus on transferring knowledge from the geometry-rich modality to the poor one in this work. Benefiting from unified feature spaces, the cross-modality transfer can be easily supported, as illustrated in Figure 4. In particular, we take features before the last ReLU layer in the voxel encoder of $\mathbf{V}_P$ as the geometry-rich teacher, marked as $\mathbf{T}_P$. Meanwhile, the feature in the same position of $\mathbf{V}_I$ is taken as the geometry-poor student, denoted as $\mathbf{S}_I$. If we take one object query position $(x,y,z)$ from Section 3.3, the feature distance for knowledge transfer is formulated as
> > >
> > > $$
> > > d_ {KT}={PL_2}(\mathbf{T}_ P(x,y,z), \mathbf{S}_ I(x,y,z)), \tag{3}
> > > $$
> > > where ${PL_2}$ represents the partial $L_2$ distance [40]. Without bells-and-whistles, the optimization objective for knowledge transfer is averaged from $N$ object queries of transformer decoder in Section 3.3, namely ${\mathcal L}_ {KT}=\frac1N\sum_i $ $(d_ {KT})$.
> > > It should be noted that the whole network is optimized in an end-to-end manner, with no need for extra procedures. Given the object position in each query, we can directly minimize the object-level distance with no need to exclude background features like [37]. In a similar pipeline, the knowledge transfer is further extended to support more input streams, like multi-frame images. The proposed cross-modality knowledge transfer is flexible with input modalities and brings consistent gains over various baselines in Tables 5 and 7.
> > >
> > > **Modality Fusion.** Different from the knowledge transfer, modality fusion aims to better utilize *all modalities* in both training and inference stages, which utilizes the complementary knowledge of point cloud and images to improve the performance and robustness. Thanks to the unified representation of each modality, feature fusion can be naturally applied. To be specific, given the processed feature space $\mathbf{V}'_I$ and $\mathbf{V}'_P$, we first select candidate modality for final prediction via modality switch, as depicted in Figure 2. That means we support single- or multi-modality input for prediction according to different settings. If both modalities are taken, $\mathbf{V}'_I$ and $\mathbf{V}'_P$ are added together to formulate the unified voxel space $\mathbf{V}_U\in{\mathbb{R}^{X\times Y\times Z\times C}}$. In this way, both modalities are well expressed in a unified manner, which can be further fused with a single convolution. The space $\mathbf{V}_U$ unifies modalities with the explicit representation, which provides an expressive space for object interactions in Section 3.3.

---

### Official Review · Reviewer_Fn2u · 2022-07-11

**Rating:** 6
**Confidence:** 4
**Soundness:** 3 good
**Presentation:** 3 good
**Contribution:** 2 fair

**Summary:**

This paper focuses on 3D object detection in driving scenarios. It proposes to fuse image features with LiDAR point cloud features following a late-fusion pipeline. The fusion leverages a learned unified 3D voxel feature space. Moreover, it adopts a transformer decoder for object relationships	modeling. A feature mimicking based method is also proposed for knowledge transfer between two input modalities. This paper demonstrates 3D detection performance improvement over single-modality and other joint-modality competing methods on the nuScenes benchmark.

**Questions:**

Please see the bullets mentioned in the weakness and limitation sections.

**Limitations:**

L-1) I’m wondering if the authors have also done experiments on comparing deformable transformer decoder with the widely adopted CenterPoint [1] detection head. This can help justify the importance and advantages of modeling object-level interactions.

L-2) Can the authors provide comparisons on the model inference speed and computation cost? We all know that latency control is critical for 3D object detection in autonomous driving scenarios.

L-3) Can the authors also provide results on other more widely adopted benchmarks such as Waymo Open Dataset and KITTI Dataset?

[1] Yin, Tianwei, Xingyi Zhou, and Philipp Krahenbuhl. "Center-based 3d object detection and tracking." Proceedings of the IEEE/CVF conference on computer vision and pattern recognition. 2021.

**Strengths And Weaknesses:**

S-1) Image and LiDAR point cloud fusion is a challenging research problem with high real-world application values.

S-2) The cross-modality interaction is simple yet effective. It could be an easily applied method for knowledge transfer between different sensors.

S-3) It is a good choice of using the transformer decoder as detection head because it can capture object-level relationships. Currently, most 3D object detection heads are based on center heat-map estimation, which lacks the ability to model interactions among different objects.

S-4) For the multi-modality fusion model training, it is a good strategy to first separately pre-train the branch of each modality and then conduct fine-tuning on the fusion model. Such a staged training pipeline can improve the training robustness and this is enabled by the proposed unified voxel feature space.

S-5) The ablation studies are extensive, which demonstrate the effectiveness of each proposed module and network architecture design. For example, Table 1 justifies the necessity of using 3D voxels instead of 2D pillars when encoding lifted image features.

S-6) Figure 5 is inspiring. It clearly demonstrates the robustness advantages of using multi-modality inputs.

W-1) In Eq. (1), how is the model trained to estimate the depth distribution $D_{I}$? For example, what losses are used? Moreover, it’s non-trivial to acquire dense depth annotation for each camera view.

W-2) For camera branch, the voxel resolution is $128 \times 128$ but $D$ is only set to 64. Thus, how do you lift camera features into the voxel space when determining the occupancy probability for voxel centers? Do you do some type of interpolation along the estimated occupancy rays?

W-3) For Table 2, how is 2D convolution used as encoders in a 3D voxel space?

W-4) In Table 5, for camera student and multi-modality teacher, where are the target/teacher features extracted from? Are they from the camera branch or the LiDAR branch, or a mixture of both?

W-5) Also in Table 5, for the best performing multi-modality fusion model, how is knowledge transfer used in its training recipe? It’s not quite clear how does knowledge transfer benefit the fused model. For example do you first train the LiDAR model, and then distill and train the camera model, which followed by the joint fine-tuning?

W-6) Does the transformer decoder jointly detect different types of objects (e.g. pedestrians, vehicles)?

---

> ### Author Response · Authors · 2022-08-02
> **Author response to Reviewer Fn2u (1/2)**
>
> Dear Reviewer *Fn2u*,
>
> Thank you for appreciating our work with valuable suggestions. We address your questions below (Q1-Q6 for the weakness part, and Q7-Q9 for the limitation part).
>
> **Q1: How the model is trained to estimate the depth distribution $\mathrm{D}_I$?**
>
> A1: We do not adopt supervision for the depth distribution $\mathrm{D}_I$. It's interesting to discover that it performs well without depth supervision in our experiments. And we find that without depth supervision, the network trends to ensure high recall of predicted depth (high activation for a range of depth) rather than a particular depth. As you mentioned that it’s non-trivial to acquire dense depth annotation, actually we attempted to use point cloud to provide sparse depth annotation that only brings 0.5% NDS gain without special design. To keep the simplicity of the proposed framework, we do not adopt depth supervision in this work. This will be made clear in the revision.
>
> **Q2: How to lift camera features with $D$ set to 64?**
>
> A2: Yes, you are right, we adopt bilinear interpolation along the estimated occupancy rays. In particular, as declared in L108-L112 of the main paper, we first project the sampling point $(x,y,z)$ to the image plane $(u,v,d)$ with the given calibration matrix $\mathbf{P}$,  where $d$ denotes the reference depth along axis $D$. Then we capture the estimated probability in distribution $\mathrm{D}_I$ using bilinear interpolation. In this manner, we can efficiently obtain the continuous depth within 64$m$ (satisfy most of the objects) from the ego vehicle. We will add more details to Section 3.1 to make this process clear.
>
> **Q3: How is 2D convolution used as encoders in the voxel space?**
>
> A3: For the Conv2D setting in Table 2 of the main paper, we process each layer of the voxel space along the axis $Z$ using 2D convolution. A simple solution is using Conv3D with kernel size $(1,3,3)$ that performs the same operation as Conv2D.
>
> **Q4: Where is the target features extracted from for multi-modality teacher?**
>
> A4: In the multi-modality knowledge transfer setting, the teacher features are extracted from the fused unified voxel space $\mathrm{V}_U$, namely the mixture of both modalities.
>
> **Q5: How is knowledge transfer used in the multi-modality model?**
>
> A5: Sorry for the confusion. Knowledge transfer and modality fusion are separate parts in the cross-modality interaction of Section 3.2. In Table 5 of the main paper, we only perform knowledge transfer from knowledge-rich settings to knowledge-poor settings, like LiDAR-based to camera-based models or multi-modality to single-modality models. For multi-modality inputs, to keep the simplicity, we optimize the whole framework in an end-to-end manner without cascade training. That means in a multi-modality setting, we do not perform knowledge transfer in the training stage. Of course, applying it in a cascade training manner may bring extra improvements. We do not use it to avoid making the pipeline complex. We will add more training details in the supplementary material to make it clear.
>
> **Q6: Does the transformer decoder jointly detect different types of objects?**
>
> A6: Yes, the transformer decoder jointly detects different objects. Actually, we also set the loss weight of different types of objects to 1.0 without special design in the training stage.
>
> **Q7: Comparisons with widely-adopted CenterPoint head.**
>
> A7: Thanks for this suggestion. We add the comparisons with the CenterPoint head with different modalities in Table A-1 of the *response to all reviewers*. Compared with the CenterPoint head, the designed transformer head achieves significant gains with 1.0% NDS and 4.1% NDS for LiDAR-based and Camera-based settings, respectively. This proves the effectiveness of the designed transformer decoder in UVTR. We will add this table to the revision.

---

> ### Author Response · Authors · 2022-08-02
> **Author response to Reviewer Fn2u (2/2)**
>
> **Q8: Provide model inference speed.**
>
> A8: We follow your suggestion and provide model inference runtime in Table A-4. As presented in the table, for the LiDAR-based setting, UVTR-L consumes cost mainly from the sparse convolution backbone. And the transformer decoder with 3 layers costs about 18ms. For the camera-based setting, the consumption is mainly from the image backbone and view transform process. The voxel encoder and transformer decoder with 6 layers also bring a noticeable cost. Here, we try to reduce the cost in the voxel encoder with spatial separate convolution. For example, we use convolution with kernel sizes $(1,3,3)$ and $(3,1,1)$ to replace Conv3D with a kernel size $(3,3,3)$ for spatial context aggregation. This choice respectively brings 5.2 ms and over 10 ms for LiDAR-based and camera-based settings without too much performance drop. In this work, we focus more on the unified representation with good performance. And the framework can be further accelerated with engineering skills. For example, we directly adopt the naive grid sampling in the view transform. It can be optimized a lot using a CUDA version operator. We will try to make the framework more efficient.
>
> Table A-4: Model inference runtime in the nuScenes *val* set. *SpaSep* indicates spatial separate convolution in the voxel encoder. We test all the models and report results on a single NVIDIA Tesla V100 GPU.
>
> | Method        | Backbone | NDS(%) | Backbone | ViewTrans | Encoder | Decoder | Total   |
> | ------------- | -------- | ------ | -------- | --------- | ------- | ------- | ------- |
> | UVTR-L        | V0.1     | 66.4   | 71.5ms   | -         | 17.1ms  | 18.4ms  | 107.0ms |
> | UVTR-L-SpaSep | V0.1     | 66.2   | 71.1ms   | -         | 11.9ms  | 18.2ms  | 101.2ms |
> | UVTR-C        | R50      | 41.9   | 103.4ms  | 64.1ms    | 32.1ms  | 36.5ms  | 236.1ms |
> | UVTR-C-SpaSep | R50      | 40.8   | 102.1ms  | 64.5ms    | 21.2ms  | 37.3ms  | 225.1ms |
> | UVTR-C        | R101     | 44.1   | 194.1ms  | 64.7ms    | 32.3ms  | 36.1ms  | 327.2ms |
> | UVTR-C-SpaSep | R101     | 43.2   | 192.1ms  | 64.6ms    | 21.9ms  | 38.2ms  | 316.8ms |
>
> **Q9: Provide results on Waymo and KITTI datasets.**
>
> A9: Thanks for this suggestion. We aim to provide a solution with the multi-view setting, and KITTI could not be so suitable for experiments. Meanwhile, compared with KITTI and Waymo datasets that use 64-beam LiDAR, the point cloud in the nuScenes is more sparse (with 32 beams) and more challenging. Thus, it could be better to perform cross-modality interaction in such a dataset. And the method that performs well in the nuScenes dataset usually achieves good results in the Waymo dataset, like CenterPoint [24]. Therefore, the detection and tracking results in the nuScenes dataset should be enough to validate the effectiveness of UVTR. Of course, following your suggestion, we try to perform experiments on the Waymo Open dataset. But such a large-scale dataset (over 1TB data) costs too many computational resources. And we cannot afford it in such a short period. We will report results in the Waymo dataset as you suggested in the final revision.

---

> ### Comment · Reviewer_Fn2u · 2022-08-08
> **The rebuttal is solid and provides answers to most of my concerns.**
>
> Thanks the authors for providing such a solid rebuttal with a lot of numbers and discussions! Most of my questions have been answered. Therefore, I increased my rating from 5 to 6.

---

> > ### Author Response · Authors · 2022-08-08
> > **Author Response**
> >
> > Dear Reviewer Fn2u,
> >
> > We sincerely thank your feedback and support. Your constructive suggestions give us guidance to greatly improve this work.

---

### Official Review · Reviewer_eNGY · 2022-07-11

**Rating:** 6
**Confidence:** 4
**Soundness:** 3 good
**Presentation:** 3 good
**Contribution:** 3 good

**Summary:**

The paper proposes a new multimodal architecture that receives images and a three-dimensional point cloud and detects 3D objects. The new architecture is based on fusing feature vectors both extracted from voxel spaces and feeding a transformer decoder with an MLP classifier. Besides being able to combine image and point cloud, experiments showed that the proposed strategy can also work in the presence of noise in the point cloud or lack of image data. Furthermore, the experimental results indicate that the proposed architecture is superior to several baselines in terms of NDS and mAP metrics.

**Questions:**

1.  It is unclear how the “sample with prob” (Figure 3) is being done. According to the text, it seems that the features are being weighted by the occupancy grid store in the tensor D. Why is this a sample and with probability?
2.  Why is FPN being used? Why partial L2 distance? Why apply deformable attention and not a vanilla attention layer? Is there any property in the task that suggests those are the better choice?
3.  The voxel encoder is shallow. Please provide more details and motivation related to the design choices.
4.  The training is not clear. Is the network training in an end-to-end manner? Are the weights of the image and voxel backbone frozen?
5.  Regarding the knowledge transfer and the training, it is not clear whether there is stage-wise training being applied or all the losses are used to train the networks in an end-to-end manner. Please kindly consider improving the description of the network training.
6.  Why is not used a linear layer or an MLP to learn how the weigh the features from the modalities before combining them?

**Limitations:**

Yes. The limitations were adequately addressed in the Discussion and Conclusion section.

**Strengths And Weaknesses:**

- Strengths:
    - While every single contribution taken separately seem limited, the integration of all idea appears to be an interesting contribution to field object detection. As far as robust multimodality fusion is concerned, the paper has shown a coherent and wise approach to provide good performance in the presence of noise and lack of data.

    - The experiments are done thoroughly, and the discussion section clearly interprets the observations from the results.

- Weakness:
    - Many important details for design choices and experiments are either missing or not properly addressed. These points are summarized and listed in the “Questions”.

    - Although the experimental results showed that the proposed approach is superior to the baselines, in several cases (multimodal approaches), the superiority is marginal, e.g., 71.1 versus 70.0, 67.1 versus 66.8.

    - Many technical components (particularly the simple ones) are adopted without strong motivation.

---

> ### Author Response · Authors · 2022-08-02
> **Author response to Reviewer eNGY**
>
> Dear Reviewer *eNGY*,
>
> Thank you for appreciating our work with valuable suggestions. We address your questions below.
>
> **Q1: Why sample with probability in view transform (Figure 3)?**
>
> A1: Thanks for this good question. Because we cannot get the real depth of each image in the camera-based setting (with the camera only). Therefore, we need to estimate the depth of each pixel when the view is transformed to the voxel space. There are actually three ways in the process **(1)** projecting each pixel like a ray with the same prob, **(2)** using estimated discrete depth, **(3)** using estimated depth distribution. For **(1)**, projecting pixels with the same prob cannot reflect the object structure in 3D space, which brings semantic ambiguity with much inferior performance in our experiments. For **(2)**, estimating discrete depth relies heavily on a pre-trained accurate depth estimator, which damages the end-to-end framework design in our UVTR. Thus, we adopt **(3)** to estimate the depth distribution $\mathrm{D}_I$ for efficient view transform, which guarantees a high recall rate in depth and can be optimized in an end-to-end manner. We will make this clear in the revision.
>
> **Q2: Design choice of FPN, partial L2 distance, and deformable attention.**
>
> A2: In general, our design aims to keep the whole framework simple for good generality. We respond to each point as follows.
>
> **(1)** Choice of FPN: To alleviate the scale variance of objects in the image plane, we adopt the widely-used FPN in 2D object detection without special design. Without FPN, the scale variance of 2D objects cannot be well handled and thus harm the aggregated feature in the voxel space.
>
> **(2)** Choice of partial L2 distance: Actually, we choose the partial L2 distance by experimental results. We compare with naive L2 distance and partial L2 distance for knowledge transfer in Table A-3. The partial L2 distance brings a 0.4% NDS gain compared with the naive version. We will add this table to the supplemental material.
>
> **(3)** Choice of deformable attention: We choose deformable attention mainly for the model efficiency. Please refer to A1 of the *response to all reviewers* for more details.
>
> Table A-3: Comparisons between L2 and partial L2 distance in the nuScenes *val* set. Models are trained in 1/4 mini nuScenes *train* set.
>
> | Method                 | Modality | Backbone | NDS(%)          | mAP(%)          |
> | ---------------------- | -------- | -------- | --------------- | --------------- |
> | UVTR-L2C-L2            | Camera   | R50      | 36.0            | 28.0            |
> | **UVTR-L2C-PartialL2** | Camera   | R50      | **36.4** (+0.4) | **28.2** (+0.2) |
>
> **Q3: More details and motivation of the shallow voxel encoder.**
>
> A3: As declared in L136-L139 of the main paper, the voxel encoder is designed to facilitate local feature interaction in the generated voxel space. Experimentally, we found that the performance of the voxel encoder saturates with 3 convolutions, and more convolutions contribute little gain (about 0.1% NDS). To save the computational cost, we set the amount to 3 by default. We will provide this detail in the revision.
>
> **Q4: Is the network trained in an end-to-end manner?**
>
> A4: Sorry for the confusion. Yes, the models with different modalities are trained in an end-to-end manner. For the multi-modality optimization, we fine-tune the backbone (not fix) that pre-trained with every single modality, as declared in L212-L2126 of the main paper. Of course, we will make this part more clear.
>
> **Q5: More details on the network training.**
>
> A5: Thanks for this suggestion! For the knowledge transfer setting, all the losses are used together to optimize the network in an end-to-end manner. In this process, to provide high-quality features for knowledge transfer, we follow the classic distillation paradigm and fix the pre-trained teacher model. We provide extra descriptions of the network optimization in the supplementary material. And we will follow your suggestion to add more descriptions for better illustration.
>
> **Q6: Using learnable weight to combine different modalities.**
>
> A6: Thanks for this suggestion! It's a good idea to dynamically combine the feature from different modalities. To keep the simplicity of the whole framework, we directly summarize the features for the unified voxel space $\mathrm{V}_U$ in the manuscript. We will try your suggestion later.

---

> > ### Comment · Reviewer_eNGY · 2022-08-09
> > **Thanks the authors for the rebuttal**
> >
> > The authors have addressed all the comments I raised in my previous review and revised the manuscript accordingly. I believe this is a solid paper that deserves to be communicated.

---

### Author Response · Authors · 2022-08-02
**Author response to all reviewers (1/2)**

Dear all reviewers,

We sincerely thank your effort in the review with valuable comments and suggestions. We first address the common concerns, followed by detailed responses to each reviewer separately. We hope our responses clarify existing concerns and make these points clear. We will really appreciate it if R3 can kindly reconsider the decision, provided that the main concerns are well addressed.

**Q1: The reason for deformable attention in the decoder** (Reviewer eNGY/Bb8v)

A1: We choose deformable attention mainly for the model efficiency.  Unlike vanilla attention that looks over all possible spatial locations for each query *Q*, the deformable attention generates sampling point $(x,y,z)$ for each object query and performs object-level interaction in a sparse manner, regardless of the spatial size of the feature maps. This is quite important, especially in a 3D voxel space with a large spatial size. Moreover, the usage of deformable attention also brings faster convergence speed during training [11]. We will make these points clear in the revision.

**Q2: Comparisons with convolution-based head** (Reviewer Fn2u/Bb8v)

A2: The transformer decoder is designed for object-level interaction and efficient object feature capture from the voxel space $\mathrm{V}_U$. Following the reviewers' suggestion, we compare it with the classic convolution-based CenterPoint head in Table A-1. In particular, to suit the CenterPoint head without bringing too much cost, we compress the axis $Z$ of the constructed unified voxel space $\mathrm{V}_U$ (after voxel encoder) in Figure 2 with a summation. As presented in Table A-1, with the same head, the proposed UVTR still surpasses CenterPoint [24] with 0.5% NDS and 1.5% mAP. Compared with the convolution-based head, the designed transformer head achieves significant gains with 1.0% NDS and 4.1% NDS for LiDAR-based and camera-based settings, respectively. This proves the effectiveness of the designed transformer decoder in UVTR.

Table A-1: Comparisons with convolution-based head in the nuScenes *val* set. *CPHead* indicates the adopted CenterPoint head.

| Method           | Modality | Backbone | Head        | NDS(%)          | mAP(%)          |
| ---------------- | -------- | -------- | ----------- | --------------- | --------------- |
| CenterPoint [24] | LiDAR    | V0.1     | Convolution | 64.9            | 56.6            |
| UVTR-L-CPHead    | LiDAR    | V0.1     | Convolution | 65.4            | 58.1            |
| **UVTR-L**       | LiDAR    | V0.1     | Transformer | **66.4** (+1.0) | **59.3** (+1.2) |
| UVTR-C-CPHead    | Camera   | R101     | Convolution | 40.0            | 35.1            |
| **UVTR-C**       | Camera   | R101     | Transformer | **44.1** (+4.1) | **36.2** (+1.1) |

**Performance on downstream tracking**

To better illustrate the capability and generality of the proposed UVTR, we further conduct experiments on the downstream tracking task. In particular, we follow the classic tracking-by-detection paradigm and apply the simple greedy tracker in CenterPoint. As presented in Table A-2, the proposed UVTR achieves leading tracking performance with the greedy tracker in different settings. Specifically, in a camera-based setting, the proposed UVTR-L2CS3 surpasses previous SOTA at the leaderboard (BEVTrack) with **17.8%** AMOTA. It further proves the effectiveness and generality of the proposed cross-modality interaction in UVTR. We will add this table to the revision.

Table A-2: Comparisons among leading methods with similar models in the nuScenes *test* set.  * indicates the state-of-the-art method at the leaderboard with no publication.

| Method           | Modality | Tracker            | AMOTA(%)         | AMOTP | Recall |
| ---------------- | -------- | ------------------ | ---------------- | ----- | ------ |
| CenterPoint [24] | LiDAR    | Greedy             | 63.8             | 0.555 | 0.675  |
| **UVTR-L**       | LiDAR    | Greedy             | **67.0** (+3.2)  | 0.656 | 0.703  |
| PolarDETR [51]   | Camera   | Transformer        | 27.3             | 1.185 | 0.404  |
| BEVTrack*        | Camera   | Private            | 34.1             | 1.107 | 0.463  |
| **UVTR-L2CS3**   | Camera   | Greedy             | **51.9** (+17.8) | 1.125 | 0.599  |
| EagerMOT [52]    | Both     | Two-stage          | 67.7             | 0.550 | 0.727  |
| AlphaTrack [53]  | Both     | Position+Apperance | 69.3             | 0.585 | 0.723  |
| **UVTR-M**       | Both     | Greedy             | **70.1** (+0.8)  | 0.686 | 0.750   |

---

### Author Response · Authors · 2022-08-02
**Author response to all reviewers (2/2)**

**Other attached experimental results**

To address the concerns of each reviewer, we also conduct the following experiments. You can find them in response to each reviewer.

- Comparisons between L2 and partial L2 distance. (Table A-3 in *Author response to Reviewer eNGY*)
- Model inference runtime. (Table A-4 in *Author response to Reviewer Fn2u*)
- Comparisons among different distances. (Table A-5 in *Author response to Reviewer Bb8v*)
- Comparisons between different weather conditions. (Table A-6 in *Author response to Reviewer Bb8v*)
- Comparisons between different light conditions. (Table A-7 in *Author response to Reviewer Bb8v*)

**Additional Reference**

[51] Shaoyu Chen, Xinggang Wang, Tianheng Cheng, Qian Zhang, Chang Huang, and Wenyu Liu. Polar Parametrization for Vision-based Surround-View 3D Detection. *arXiv:* 2206.10965, 2022.

[52] Aleksandr Kim, Aljovsa Ovsep, and Laura Leal-Taixe. EagerMOT: 3D Multi-Object Tracking via Sensor Fusion. In *ICRA*, 2021.

[53] Yihan Zeng, Chao Ma, Ming Zhu, Zhiming Fan, and Xiaokang Yang. Cross-Modal 3D Object Detection and Tracking for Auto-Driving. In *IROS*, 2021.

---

### Meta-Review · Area_Chair_kQz1 · 2022-08-26

**Recommendation:** Accept
**Confidence:** Less certain

**Metareview:**

The paper proposes a multimodal system for 3d object detection and 3 expert reviewers vote for its acceptance, after rebuttal, based on their appreciation of the good improvements brought by multimodality, and due various interesting details of the system.

I agree with reviewer Bb8v that the writing should be polished, starting from the abstract, e.g. "Benefit from the unified manner, cross-modality interaction is then proposed to make full use of inherent properties from different sensors" -- this sentence reads very poorly.

**Award:**

No

---

### Decision · Program_Chairs · 2022-09-14

Accept